# Imaging Peripheral Nerves In Vivo with CT Neurogram Using Novel 2,4,6-Tri-Iodinated Lidocaine Contrast Agent

**DOI:** 10.3390/bioengineering12040422

**Published:** 2025-04-16

**Authors:** Rui Tang, Ron Perez, David M. Brogan, Mikhail Y. Berezin, James E. McCarthy

**Affiliations:** 1Department of Radiology, School of Medicine, Washington University, St. Louis, MO 63110, USA; rtang@wustl.edu; 2Department of Orthopedic Surgery, School of Medicine, Washington University, St. Louis, MO 63110, USA; perezron@wustl.edu (R.P.); brogand@wustl.edu (D.M.B.); 3MacHouse Designs LLC, Loveland, CO 85039, USA

**Keywords:** nerve imaging, nerve, neurography

## Abstract

Peripheral nerve injuries are a significant concern in surgical procedures, often leading to chronic pain and functional impairment. Despite advancements in imaging, preoperative and intraoperative visualization of peripheral nerves remains a challenge. This study introduces and evaluates a novel tri-iodinated lidocaine-based contrast agent for computed tomography neurography, aiming to enhance the intraoperative visibility of peripheral nerves in vivo. A tri-iodinated lidocaine analogue was synthesized and characterized for its radiodensity, sodium channel binding and nerve affinity. Sodium channel affinity was performed using molecular docking. In vitro contrast enhancement was assessed by comparing the agent’s Hounsfield unit (HU) values with those of Omnipaque, a clinically approved contrast medium. In vivo imaging was conducted on rat sciatic nerves using micro-CT, followed by ex vivo validation. Nerve conduction blockade was assessed via electrical stimulation and histological analysis was performed to evaluate neurotoxicity. Experimental results revealed the tri-iodinated lidocaine analogue to have similar or higher affinity toward voltage-gated sodium channels than the parent lidocaine and a radiodensity comparable to the commercial CT contrast agent Omnipaque in vitro. In vivo, the contrast agent provided CT visualization of the sciatic nerve, with a significant increase in HU values compared to untreated nerves. Electrical stimulation confirmed transient nerve conduction blockade without observable histological damage, supporting its dual role as an imaging and nerve-blocking agent. This study presents a novel tri-iodinated lidocaine-based contrast agent that enables clear CT visualization of peripheral nerves while maintaining reversible nerve inhibition. These findings support its potential application in preoperative planning and intraoperative nerve protection to reduce surgical nerve injuries. Further studies are warranted to optimize imaging conditions and evaluate its clinical feasibility.

## 1. Introduction

Peripheral nerve injuries are a significant risk in surgical procedures, with potentially severe consequences such as chronic pain, loss of motor function, and compromised quality of life. Because the majority of operations pose some risk of nerve injury, it is difficult to generalize the risk of nerve injury during surgery. The likelihood of nerve injury varies depending on the type of procedure, patient-specific factors, and intraoperative conditions. For example, in parotidectomy, the frequency of nerve injury varies between 37.5% and 62.5% [1]. Iatrogenic nerve injuries are a major concern and can result from direct surgical trauma, mechanical stress due to improper positioning, or other mechanisms. The incidence of significant peripheral nerve injury has been reported to be 0.2% to 3.7% in total hip arthroplasty and 0.16% to 1.5% in total knee arthroplasty [2]. The incidence of nerve injury in hip arthroscopy ranges from 1.4% to 5%, with the pudendal and lateral femoral cutaneous nerves being most commonly affected [3]. Other procedures with a risk of nerve injury include lymph node dissections, tumor resections, and inner ear surgeries, among others [4]. Overall, the incidence of perioperative peripheral nerve injury is generally less than 1% in the general surgical population [5]. Using a conservative estimate of 0.5% for all surgical procedures with 40 million surgeries performed annually in the United States, this would result in approximately 200,000 nerve injuries annually.

Despite advances in surgical imaging and preventive measures, thorough anatomical knowledge, visual identification of at-risk nerves, intraoperative nerve monitoring, and meticulous surgical techniques remain a challenge [4,5,6]. Currently, most surgeries rely heavily on the experience and anatomical knowledge of the surgeon to avoid nerve damage, as existing imaging technologies often fail to provide clear, preoperative or intraoperative visualization of peripheral nerves. It is notable that surgical navigation using CT imaging is now an embedded infrastructure within medicine. However, surgical navigation primarily exists for neurologic tumors, spine surgery, joint replacement, craniofacial reconstruction, and interventional radiology, leaving the surgical navigation of peripheral nerves in uncharted waters [7]. Developing a reliable clinically useful tool to provide nerve visualization using ionizing radiation (e.g., CT) would permit preoperative or intraoperative mapping to avoid nerve damage during procedures, potentially reducing relevant and meaningful postoperative complications [8].

To address this clinical need, several advanced techniques to visualize peripheral nerves during surgery, with and without contrast agents, have been proposed. Nerve autofluorescence under near-ultraviolet light has been applied to visualize superficial nerves intraoperatively [1,9,10]. Diffuse reflectance spectroscopy that differentiates nerves from surrounding tissues based on their higher photon scattering and differences in oxygenated hemoglobin has also been explored as a label-free method for intraoperative nerve detection [11]. Polarization-sensitive optical coherence tomography (OCT) offered the potential to visualize nerves with axon-scale resolution, though remains in development [12]. Recently, a fluorescence-guided imaging method with nerve-specific fluorophores that enhances the visibility of nerves against surrounding tissues has been reported [13,14]. However, most of the optical techniques suffer from low tissue depth penetration. Photons in the visible and near-infrared wavelengths tend to scatter in tissue, at the expense of the spatial resolution that is required for the identification of nerves with high precision.

Recently, our team has developed a new approach for imaging nerves using a lidocaine scaffold carrying an iodine atom, the union of which targets the voltage-gated sodium channel (NaV) in nerves and provides absorption of X-rays used in CT imaging [15]. In this work, we further optimized the imaging agent by introducing three iodine moieties into the lidocaine template. The three-iodine substitution significantly increased the signal-to-noise ratio while still retaining the capability of binding to nerves.

## 2. Methods and Materials

### 2.1. Synthesis of a 2,4,6, Triiodo-Substituted Lidocaine

The imaging agent 2,4,6 triiodo-substituted lidocaine [3] was prepared via a two-step procedure shown in Figure 1.

Step 1: To a solution of 0.35 g of 2,4,6-triiodo-phenylamine **1** (CAS 24154-37-8, Santa Cruz Biotechnology, Dallas, TX, USA) (0.74 mmol, 1.0 eq.) in 10 mL of dichloromethane **1** (DCM, ≥99.5%, CAS 75-09-2, Sigma Aldrich, St. Louis, MO, USA), triethylamine (TEA, 0.12 mL, 1.2 eq., Sigma Aldrich, St. Louis, MO, USA) was added. The solution was cooled to 4 °C and chloroacetyl chloride (0.07 mL, 1.2 eq., 98%, CAS 79-04-9, Santa Cruz, Dallas, TX, USA) at 4 °C was cooled dropwise on ice. The reaction mixture was warmed to room temperature under stirring and stirred for 4 h. After the completion of the reaction, water (5 mL, MilliQ) was added, and the mixture was extracted with DCM (3×). The combined DCM fractions were dried over sodium sulfate concentrated under vacuum. The crude product was purified by recrystallization from ethanol/water (1:1) to yield 0.273 g of (2,4,6-triiodophenyl)-2-chloroacetamide **2**, which was used in Step 2 without additional purification and analysis.

Step 2: Intermediate **2** (0.27 g, 0.49 mmol, 1.0 eq.) and sodium iodide (0.369 g, 5.0 eq. ≥99.5%, CAS 7681-82-5, Sigma Aldrich, St. Louis, MO, USA) were dissolved in acetonitrile (5.0 mL, 99.8%, CAS 75-05-8, Sigma Aldrich, St. Louis, MO, USA) at room temperature. To the stirred solution, diethylamine (0.255 mL, 5.0 eq. ≥ 99.5%, CAS 109-89-7, Sigma Aldrich, St. Louis, MO, USA) was added and the reaction mixture was stirred overnight. The mixture was subsequently concentrated and the residue portioned between DCM and water. The organic phase was dried and concentrated to a crude solid. Crude material was purified by silica gel column chromatography to obtain the target product **3** (0.117 g, 40.9% yield, 98.8% LC/MS purity at 254 nm). The confirmation of product **3** is provided in the Appendix A.

The substance peak was detected using high-performance liquid chromatography (HPLC) coupled with mass spectrometry (LC-MS). The UV detector in the HPLC was set to 254 nm, a wavelength at which most organic compounds exhibit absorption. A strong absorption peak and mass spectrometry signal were observed at approximately the same retention time (3.6–3.3 min). The identity of the peak was further confirmed by analyzing the mass spectrum across the peak.

LCMS: Thermo-Fisher, Agilent Pursuit C18 2 × 50 mm; flow rate 0.5 mL/min, UV: 254 nm, solvent 5–95%, MeCN (0.1% formic acid) in H_2_O (0.1% formic acid) over 10 min, Rf = 3.63 min; MS: (ESI positive ion mode) *m*/*z* 584.61 (M + 1, MW = 583.832). Peak purity = 98.8%. Prepared batch: 116.5 g.

^1^H NMR: (300 MHz, MeOD) δ8.24 (s, 2 H), 3.24 (s, 2 H), 2.75 (q, 4 H), 1.16 (t, 6 H). A NH proton would normally be expected in the ^1^H NMR spectrum. However, the spectrum was acquired in methanol-d_4_ (MeOD), which readily exchanges labile protons. As a result, the NH proton undergoes rapid exchange with deuterium from the solvent and becomes N–D, making it invisible in the ^1^H NMR spectrum.

### 2.2. Molecular Docking

Molecular docking was conducted using PyRx Virtual Screening Tool software (Version 1.1, SourceForge, San Diego, CA, USA) employing the AutoDock Vina algorithm. The 3D structure of the NavAb voltage-gated sodium channel transport protein in a closed conformation (PDB: 5VB2) [16] was downloaded from the Protein Data Bank (https://www.rcsb.org/ (accessed on 11 April 2025)). The structure of the protein was first prepared by removing water molecules and removing a docked ligand with it. Lidocaine and triiodo-lidocaine structures were optimized and converted to PDBQT format via the Chem3D package (Version 23.1.2, Revvity Signals Software, Waltham, MA, USA). A full grid was defined for docking, encompassing the entire structure of the transport protein to allow for the identification of potential binding sites without predefined bias. The docking results were analyzed for binding energy, and the top-ranked conformations were visualized using Discovery Studio for detailed interaction mapping in 2D and PyMOL (Version 2.3.0, Schrödinger, Inc., New York, NY, USA) for 3D representation.

### 2.3. Contrast Testing In Vitro

2,4,6-triiodine-substituted lidocaine analogue was dissolved in ethanol at 25 mg/mL and iohexol (trade name Omnipaque, GE Healthcare, Chicago, IL, USA) was dissolved in deionized water (25 mg/mL). Iohexol is a nonionic, water-soluble radiographic contrast medium widely used to enhance CT image contrast, improving the visualization of blood vessels, organs, and other tissues [17]. The equal volumes of solution were placed within optically transparent 3 mm tubes (OD) (iThera Medical GmbH, München, Germany). The tubes were imaged within an IVIS SpectrumCT (Revvity, Waltham, MA, USA) using the following settings: dose range: 13 mGy, X-ray energy: 50 kVp, resolution: 150 µm, scan time: 90 s per scan. Measurements of the Hounsfield unit (HU) density were then taken for equivalent surface areas using the entire cross-section of the tube in the axial plane.

### 2.4. Animal Models

All animal experiments were conducted in compliance with Washington University Institutional Animal Studies Committee and NIH guidelines. Three Lewis rats (RRID:RGD_737932) aged 8–12 weeks old (Charles River Laboratory, Wilmington, MA, USA) were used in this study. Animals were housed with free access to water and food and maintained on a 12:12 h light/dark cycle with controlled temperature and humidity. Prior to the surgery, rats were anesthetized using inhalational isoflurane at a rate of 2 L/min and the designated hindlimbs were shaved. After the CT imaging and electrical stimulation, the rats were euthanized with a carbon dioxide overdose.

### 2.5. In Vivo Imaging Procedure of Exposed Nerve

The skin of the rat over the sciatic nerve was excised and a transgluteal approach was performed in an atraumatic fashion by using microsurgical instrumentation and neurolysis of both sciatic nerves to expose a length of 3 cm. The contrast agent was applied to the right side (1 mL, 25 mg/mL) and then aspirated after 10 min, while the left sciatic nerve received no contrast (Figure 1A). The rat was then imaged using a nanoScan PET/CT (Siemens, München, Germany) (Figure 1B) with X-ray power: 50 kVp × 980 μA, computed tomography dose index (CTDI): 30 mGy, dose length product (DLP): 582.6 mGy × cm. The exposure time was 170 ms, with the total exposure being 0.17 mAs. The images were analyzed using Inveon Research Workspace software (Version 4.2, Siemens, München, Germany).

### 2.6. Electrical Stimulation

Nerve conduction was then investigated in the sciatic nerve receiving the contrast and the contralateral untreated sciatic nerve by applying a light pinch in each animal with repetitive grasping with a micro forceps for 10 s and with application of an electrical impulse using a nerve stimulator (Checkpoint Surgical, Inc., Independence, OH, USA) on continuous 150 μs pulse 0.5 mA for 10 s, while monitoring for limb movement (Figure 2). The presence or absence of paw plantarflexion reflected the paralyzing effect of the lidocaine derivative.

### 2.7. Imaging of Sciatic Nerves Ex Vivo

After euthanasia, sciatic nerves from both sides were explanted, rinsed with saline and placed in empty 1.5 mL Eppendorf tubes. The tubes were placed in the NanoScan PET/CT and imaged under the same imaging conditions as mentioned above (see In Vivo Imaging in Section 2). The Hounsfield unit densities were analyzed with Inveon Research Workspace software (Version 4.2, Siemens, München, Germany). The ex vivo sciatic nerve specimens and contralateral negative controls were sectioned, mounted and stained with OsO_4_ and toluidine blue as previously described [19], and inspected via light microscopy for evidence of axon damage.

## 3. Results

Optical transparent clear straws filled with the solutions of triiodo-lidocaine in ethanol and Omnipaque in water embedded in agar phantom (iThera Medical, Munich, Germany) showed the same level of Hounsfield unit (HU) density from the regions of interest 745.0 ± 9.0 HU and 755.0 ± 8.5 HU, respectively (Figure 3).

The molecular docking simulation using AutoDock Vina successfully predicted the binding of lidocaine and triiodo-lidocaine to the voltage-gated sodium channel (NavAb). AutoDock Vina employs the Broyden–Fletcher–Goldfarb–Shanno (BFGS) optimization algorithm for energy minimization, automatically detecting rotatable bonds in the ligand while assuming a rigid receptor [20]. The scoring function, which approximates the binding free energy, considers van der Waals forces, hydrogen bonding, electrostatics, and desolvation effects.

The docking results indicate that both lidocaine and triiodo-lidocaine preferentially bind within the central cavity, interacting strongly with S6 transmembrane helices—key regions for channel gating (Figure 4A,B). The lowest-energy conformations confirm that these molecules sterically hinder ion passage, supporting their role as sodium channel blockers. Lidocaine exhibited a binding energy of −5.8 kcal/mol, while triiodo-lidocaine showed −6.8 kcal/mol, suggesting a similar moderate but slightly enhanced interaction for the latter.

Figure 4 presents the 3D molecular structure of NavAb in a closed conformation, with lidocaine and triiodo-lidocaine positioned inside the pore. The tetrameric assembly of the sodium channel is evident, with red helices representing transmembrane domains spanning the lipid bilayer (Figure 4A). Both compounds are located within the central pore, interacting with pore-lining residues, effectively blocking ion conduction. This aligns with lidocaine’s known anesthetic mechanism, which prevents neuronal depolarization and action potential propagation.

The docking results indicate that both lidocaine and the 2,4,6-triiodine-substituted lidocaine analogue preferentially bind inside the central cavity. The lowest-energy binding configurations, identified through AutoDock Vina’s scoring function, showed strong interactions with residues lining the S6 transmembrane helices (Figure 4A,B,E,F), which are critical for channel gating. Visualizations of the docked complex confirm that lidocaine adopts a conformation that sterically hinders ion passage. Overall, these findings support the mechanism by which triiodine lidocaine exerts its anesthetic effects, as demonstrated below.

The sciatic nerves exposed to the nerve contrast demonstrated in vivo enhancement with microCT imaging (Figure 5A) with an average signal intensity of 525 HU. For comparison, bone is around ~1500 HU and other soft tissues are <50 HU. The negative control had no evidence of in vivo nerve enhancement (0 HU).

### Peripheral Nerve Electrical Stimulation

In our study, we used a peripheral nerve stimulator (Checkpoint Surgical) set at 150 μs pulse width to apply direct electrical stimulation to the sciatic nerve in a rat. On the control side, stimulation elicited a robust muscle twitch in the paw (Appendix A), confirming normal nerve conduction and neuromuscular function. In contrast, on the treated side, where the sciatic nerve was exposed to triiodo-lidocaine, no muscle twitching was observed. This lack of response suggests that the lidocaine derivative effectively blocked nerve conduction by inhibiting voltage-gated sodium channels, as demonstrated with the docking experiments, preventing action potential propagation along the motor fibers. This outcome demonstrates the compound’s ability to retain its binding affinity for sodium channels in the nerve, making it a potential candidate not only for imaging but also for temporary nerve inhibition in clinical applications.

The affinity of the contrast agent was further confirmed ex vivo by extracting the sciatic nerves from the euthanized rats. These were washed, inserted in small Eppendorf tubes and placed on a nanoScan PET/CT scanner. The image shown in Figure 5B clearly shows a strong signal from the treated nerve.

On gross inspection after the 10 min dwell time of the nerve contrast, none of the rats demonstrated evidence of soft tissue injury or injury to the nerve. Following the experiment, the sciatic nerves from both the control and lidocaine derivative-treated sides were excised for histological analysis. The nerves were processed with osmium tetroxide (OsO_4_) staining to highlight myelin integrity and further counterstained with toluidine blue for detailed morphological assessment. Histomorphometry analysis of the images (axon density, myelination level of axons, G-ratio; see the representative image in Figure 6) revealed no significant structural abnormalities in either group, indicating that despite the functional nerve blockade observed on the treated side, the underlying nerve architecture remained intact. This suggests that the triiodine lidocaine effectively inhibits conduction without causing detectable histological damage to the nerve, supporting its potential as a reversible nerve-blocking agent without inducing acute neurotoxicity and damage to the nerve.

## 4. Discussion

Since its development in 1943 [21], lidocaine, also known by its brand name Xylocaine, has been one of the most commonly used local anesthesia agents for surgical procedures. Lidocaine alters neuronal signal transmission by prolonging the inactivation of fast voltage-gated sodium channels in the cell membrane of neurons, which are responsible for action potential propagation [22,23]. The combination of lidocaine’s high affinity for nerves and low neurotoxicity has led to new interest in its potential application as an integral component in imaging contrast agents. For example, a recent study demonstrated that 18F-labeled lidocaine can be used as a nerve-specific PET imaging agent [24].

The lidocaine molecule is composed of three key structural components (domains), each playing a crucial role in its function as a local anesthetic and nerve-blocking agent: a hydrophilic domain, linker and lipophilic domain (Figure 7). The hydrophilic domain is represented by a tertiary amine and is responsible for interacting with the intracellular binding site of sodium channels. In its protonated form, it physically blocks the Na^+^ ion conduction pathway, preventing depolarization and action potential propagation [25]. The ability of lidocaine to exist in both charged and uncharged states allows it to diffuse through the membrane in its neutral form and then become protonated in the cytoplasm, enabling effective sodium channel blockade. The linker domain connects the aromatic ring to the tertiary amine and determines the metabolic stability of the molecule. Lidocaine belongs to the amide-type local anesthetics, which are metabolized primarily by the liver (cytochrome P450 enzymes) [26], in contrast to ester-linked anesthetics, such as procaine and tetracaine, which are hydrolyzed more rapidly by plasma esterases [27]. The lipophilic region is essential for membrane permeability and channel binding affinity. It facilitates the penetration of lidocaine into the nerve cell membrane, allowing it to reach the central cavity of sodium channels [28]. This tripartite structure enables lidocaine and its derivatives to effectively modulate nerve excitability, making it useful for local anesthesia and nerve blocks. We rationalized that modifications to the lidocaine structure by introducing iodines to the hydrophobic domain would not change the specificity of the molecule and would retain its affinity while providing the necessary contrast. The rationale was confirmed by molecular docking studies showing that the triiodo-lidocaine occupies the same location withing the pore of the model voltage-gated sodium channel NavAb.

Iodine has a strong attenuating effect on X-rays primarily due to its high atomic number (Z = 53), which enhances the photoelectric (Compton) effect. When X-rays interact with iodine atoms, the probability of photoelectric absorption increases significantly, leading to a substantial increase in X-ray attenuation. Additionally, the high atomic number of iodine offers a large number of electrons, which increases the probability of interactions with incoming X-ray photons. This results in greater absorption and attenuation of the X-ray beam, making iodine an effective contrast agent in radiographic imaging [29].

To increase the CT contrast of the nerves, our team has previously developed a mono-iodinated lidocaine (Figure 7) by attaching iodine to the lipophilic domain of lidocaine [15]. In vitro testing of this compound showed significantly increased (11.6 fold, *p* < 0.001) radiodensity compared to lidocaine without conjugated iodine. The compound was also functional, producing a paresis of the hindlimb.

However, mono-iodinated lidocaine was insufficient for in vivo peripheral nerve enhancement with CT [15] due to its lower molar concentration of contrast-generating iodine. This led us to develop the 2,4,6-tri-iodinated lidocaine derivative. Given the CT signal is approximately linearly proportional to the concentration of iodine within a certain range, the new compound was expected to triple the signal and improve the signal-to-noise ratio.

The results of this pilot study support our central hypothesis that an iodinated local anesthetic can provide in vivo contrast nerve enhancement with the use of a CT scanner. Iodinated contrast agents are the most frequently used contrast agents globally, mostly within the vascular system. Given the findings in this study demonstrating an in vivo density of 300 HU and ex vivo density of 575 HU, we anticipate that clinically meaningful and precise radiographic data may be obtained using CT neurography.

The potential advantages over the proposed approach include the enhancement of nerves at all tissue depths, 3D visualization, and the mapping of peripheral nerve anatomy prior to incision. CT offers high spatial resolution, ranging from 0.5 to 1 mm for standard CT and as high as 0.1 mm for High-Resolution CT (HRCT). Intraoperative CT [30] systems have emerged in the last decade as an adjunct to surgical navigation for real-time spatial resolution of intraoperative anatomy, particularly in spine procedures. Therefore, a clinical imaging agent capable of enhancing neural anatomy on CT scans may find ready applications in high-risk procedures, particularly if it is compatible with existing intraoperative imaging modalities.

The visualization of nerves, such as the brachial plexus, typically requires high contrast densities to achieve adequate image quality and delineation [31,32]. The study by Claves et al. on the evaluation of contrast densities for CT angiography found that a contrast density of 150 HU was optimal for measuring carotid stenosis [32]. Given the higher complexity and smaller size of neural structures compared to vascular structures, higher contrast densities are likely required for adequate visualization. The synthesis of other iodinated local anesthetics and optimization of imaging conditions, as well as post-processing techniques to achieve higher contrast, will form the basis of future work.

Intraoperative CT is routinely utilized during spine surgery to evaluate the location of pedicle screws and register bony and anatomic landmarks for intraoperative navigation. Furthermore, surgical navigation using preoperative CT to identify relevant anatomy during a surgery combined with a stylus registered to the preoperative CT has been used in neurosurgery for tumor removal and otorhinolaryngology for 20 years, becoming an industry standard of care. Thus, the ability to use CT for either preoperative or intraoperative navigation by permitting the nerve to be visualized using X-rays offers the potential to assist with nerve root identification and registration when utilizing intraoperative CT.

## 5. Conclusions

Perioperative nerve injuries represent a substantial clinical challenge with significant consequences for patients. Given the high incidence of nerve injuries across a wide range of surgical procedures, incorporating direct visualization techniques into routine surgical practice could play a crucial role in reducing iatrogenic nerve damage. Our proposed novel tri-iodinated lidocaine analogue retains the binding properties of the parent lidocaine while being a viable nerve CT imaging agent. The probe binds to the nerve-specific sodium channels and lidocaine’s receptors, causing paresis without cellular injury. Such technology has the potential to enhance surgical safety, improve patient outcomes, and decrease the long-term burden of surgical nerve injuries. The clinical applications of our study extend beyond surgical nerve visualization to potential use in pain management, intraoperative nerve preservation, and interventional procedures such as regional anesthesia and nerve-targeted drug delivery. Further translational studies will be focused on optimizing imaging parameters, the evaluation of long-term safety, and integration into existing surgical protocols.

## Data Availability

The data presented herein can be found at DOI: 10.17632/7h7cysgjsx.1; video recordings demonstrating the nerve conduction blockade effect can be found at the following link: https://youtube.com/shorts/R5E4vxFlqJM (accessed on 11 April 2025).

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
