# Peer review of "Imaging Peripheral Nerves In Vivo with CT Neurogram Using Novel 2,4,6-Tri-Iodinated Lidocaine Contrast Agent"

_bioengineering, 2025, doi:10.3390/bioengineering12040422_

Round 1

Reviewer 1 Report

Comments and Suggestions for Authors

The paper "Imaging Peripheral Nerves In Vivo with CT Neurogram Using Novel 2,4,6-Triiodinated Lidocaine Contrast Agent" is interesting and well presented overall; however, some improvement is necessary before it can be published. 
Point 1. Please explain what does (1) in abstract strand for (Line 15). 
Point 2. Please check the magnification. Pictures D and H do not look like they are under the same magnification. Please correct it.
Point 3. Please correct the typos in the text (for example Figure 6) line 272). 
Point 4. Please move Figure 7 out of the discussion section. You can move it to the material and methods or supplement section.
Point 5. How many rats did you include in this study? Please add it to the part devoted to animal models.

Author Response

REVIEWER 1

Point 1. Please explain what does (1) in abstract strand for (Line 15).

Response:  We thank the reviewer for catching a typo. We removed (1) from the abstract.

Point 2. Please check the magnification. Pictures D and H do not look like they are under the same magnification. Please correct it.

Response: Thank you for the important suggestion. The PyMol software that we used to process the images does not have a function to select the view under the same magnification. However, we were able to manually zoom in both ROIs to the same scale. Figure 4 is updated accordingly as requested.

Point 3. Please correct the typos in the text (for example Figure 6) line 272). 

Response: We corrected the typos.

Point 4. Please move Figure 7 out of the discussion section. You can move it to the material and methods or supplement section.

Response: Figure 7 illustrates the concept of retaining the active part of the lidocaine molecule and the linker while modifying the inactive part. Per Reviewer 2 request we have modified this figure.

Point 5. How many rats did you include in this study? Please add it to the part devoted to animal models.

Response: We thank the reviewer for pointing to this question. We used three rats. We added this information in the Methods section.

Reviewer 2 Report

Comments and Suggestions for Authors

I was glad to read this article, devoted to a very important topic of advanced approaches to nerve imaging. It is important that this work is a study that describes not only the synthesis of a new effective contrast agent (moreover, one that exhibits a functional role), but also its application in the corresponding in vivo studies on nerve tissue imaging.

Nevertheless, I think it is important to note some points, the work on which can improve the quality of this material.

The introduction, among other issues, discusses alternative methods of nerve imaging, indicates some of their disadvantages, but does not highlight certain advantages compared to the CT method described in the article, the main one of which is the possibility of imaging during surgery, which is impossible when using CT.

Section 2.1 discusses the details of obtaining and identifying the intermediate and target compounds. It is worthwhile to give uniformity to the text and indicate for all commercially available substances the CAS data (this one is not necessary), the manufacturer and the degree of purity of the compound.

The LCMS method was used for identification, but for some reason the wavelength of UV light absorption is indicated, which is used in the HPLC method with spectrophotometric detection. Question: how was the substance peak detected on the chromatogram? Mass-spectrometrically or spectrophotometrically? Depending on the answer, it is worthwhile to bring the data in the article into line.

In this section, as in some other places in the article, there are minor typos, such as the absence of spaces, double spaces, etc. But there are also more important errors in this place, such as the spelling of H2O and nM (nm is correct!).

The 1H NMR data for compound 3 are provided. Due to the simplicity of its spectrum, I think it is not superfluous to provide the assignment of the spectrum (to indicate which signals correspond to which hydrogen atoms). (And to mention, why the signal of the NH group does not appear.)

Since compound 2 is also new, it is worthwhile to provide its identification data (MS and NMR). Otherwise, questions arise as to whether this substance was really isolated in pure form. It would be also better to provide 13C NMR spectra for compounds 2 and 3, since this is a standard in describing new compounds.

The section contains a reference to supporting materials file, but it was not available (at least to me as a reviewer). Please note this or be sure to add these data if was not provided.

Note the repetition of technical details in the captions to most figures, which are displayed mostly unchanged in the main text right before the figures. I think such repetition may be unnecessary.

In section 2.3. the X-ray dose is indicated in mG, then as mGy, in section 2.5. the current is indicated as uA, then uAs (which is more correct). It is worth providing these data uniformly.

Figure 7 on the right shows the structures of the diiodinated analogue of lidocaine and an alternative triiodinated derivative. But there are no references to these compounds and no discussion of them in the main text. Why are they provided? It is worth either removing them or providing a description and discussion.

Overall, I believe that this work has good potential, requires only some edits and, accordingly, can be published after a minor revision.

Author Response

REVIEWER 2

Comment 1. The introduction, among other issues, discusses alternative methods of nerve imaging, indicates some of their disadvantages, but does not highlight certain advantages compared to the CT method described in the article, the main one of which is the possibility of imaging during surgery, which is impossible when using CT

Response: We express our gratitude to the reviewer for their perspective and the opportunity to provide additional information that may address the reviewer’s concern. Specifically, we added the following paragraph to the Discussion section:

Intra-operative CT is routinely utilized during spine surgery to evaluate location of pedicle screws and register bony and anatomic landmarks for intra-operative navigation.  Furthermore, surgical navigation using preoperative CT to identify relevant anatomy during a surgery combined with a stylus registered to the preoperative CT has been used in neurosurgery for tumor removal, and otorhinolaryngology for 20 years becoming an industry standard-of-care. The ability to thus use CT for either preoperative or intra-operative navigation by permitted the nerve to be visualized using x-ray offers the potential to assist with nerve root identification and registration utilizing intra-operative CT. 

Comment 2. Section 2.1 discusses the details of obtaining and identifying the intermediate and target compounds. It is worthwhile to give uniformity to the text and indicate for all commercially available substances the CAS data (this one is not necessary), the manufacturer and the degree of purity of the compound.

Response: The reviewer’s observations and recommendation are received and appreciated. We have added the manufacturer and CAS description where lacking and have added or confirmed chemical purity where needed in the Method Section.

Comment 3. The LCMS method was used for identification, but for some reason the wavelength of UV light absorption is indicated, which is used in the HPLC method with spectrophotometric detection. Question: how was the substance peak detected on the chromatogram? Mass-spectrometrically or spectrophotometrically? Depending on the answer, it is worthwhile to bring the data in the article into line.

In this section, as in some other places in the article, there are minor typos, such as the absence of spaces, double spaces, etc. But there are also more important errors in this place, such as the spelling of H2O and nM (nm is correct!).

Response: We thank the reviewer for the opportunity to clarify the analytical method and added the following clarification to the manuscript: The substance peak was detected using high-performance liquid chromatography (HPLC) coupled with mass spectrometry (LC-MS). The UV detector in the HPLC was set to 254 nm, a wavelength at which most organic compounds exhibit absorption. A strong absorption peak and mass spectrometry signal was observed at approximately the same retention time. The identity of the peak was further confirmed by analyzing the mass spectrum across the peak.

We thank the reviewer for catching these minor but meaningful errors which have been subsequently reconciled.

Comment 4. The 1H NMR data for compound 3 are provided. Due to the simplicity of its spectrum, I think it is not superfluous to provide the assignment of the spectrum (to indicate which signals correspond to which hydrogen atoms). (And to mention, why the signal of the NH group does not appear.)

Response: We thank the reviewer for the suggestion and agree that the ¹H NMR spectrum is relatively simple. However, we would like to keep the assignment of the protons to demonstrate the purity of the final product and to provide clarity and reproducibility for future reference, especially for readers who may use this compound in subsequent studies.

We also agree with the reviewer that an NH proton would normally be expected in the ¹H NMR spectrum. However, the spectrum was acquired in methanol‑dâ‚„ (MeOD), which readily exchanges labile protons. As a result, the NH proton undergoes rapid exchange with deuterium from the solvent and becomes N–D, making it invisible in the ¹H NMR spectrum. This is a well‑known effect for NH and OH protons in protic deuterated solvents such as MeOD. We added the following clarification to the Methods and Materials section: NH proton undergoes rapid exchange with deuterium from the solvent and becomes N–D, making it invisible in the ¹H NMR spectrum.

Comment 5: Since compound 2 is also new, it is worthwhile to provide its identification data (MS and NMR). Otherwise, questions arise as to whether this substance was really isolated in pure form. It would be also better to provide 13C NMR spectra for compounds 2 and 3, since this is a standard in describing new compounds

Response: we thank the reviewer for these insightful comments. Indeed, the compound 2 was not isolated in its pure form and after recrystallization was used in the second step reaction without additional analysis. We added this clarification in the Methods and Materials section: The crude product was purified by recrystallization from Ethanol/water (1:1) to yield 0.273 g of (2,4,6-triiodophenyl)-2-chloroacetamide 2 that was used in Step 2 without additional purification and analysis.

We also agree that while 13C NMR is a standard method, we relied on two orthogonal methods 1H NMR and mass-spec to validate the structure.

Comment 6. The section contains a reference to supporting materials file, but it was not available (at least to me as a reviewer). Please note this or be sure to add these data if was not provided.

Response: We apologize for the confusion. Our supplemental Information includes the complete analysis of the Compound 3 with HPLC, MS and 1H NMR spectra and the Supplemental Movie showing electrical stimulation of the sciatic nerves. 

Comment 7. Note the repetition of technical details in the captions to most figures, which are displayed mostly unchanged in the main text right before the figures. I think such repetition may be unnecessary.

Response: we thank the reviewer for this valuable suggestion. We modified the captions to avoid duplications.

Comment 8. In section 2.3. the X-ray dose is indicated in mG, then as mGy, in section 2.5. the current is indicated as uA, then uAs (which is more correct). It is worth providing these data uniformly.

Response: We thank the reviewer for their exquisite attention to detail toward this recommendation. The error in excluding the ‘y’ in ‘mGy’ has been remedied in keeping with the reviewer’s keen observation.

Comment 9. Figure 7 on the right shows the structures of the diiodinated analogue of lidocaine and an alternative triiodinated derivative. But there are no references to these compounds and no discussion of them in the main text. Why are they provided? It is worth either removing them or providing a description and discussion

Response: We again express our gratitude to the reviewer for the opportunity to make our manuscript more pointed and clearer. The referenced image has been modified to focus on the contrast agent listed, 2,4,6-Tri-iodinated lidocaine analogue.

Comment 10: Overall, I believe that this work has good potential, requires only some edits and, accordingly, can be published after a minor revision.

Response: we thank the reviewer for the positive feedback

Round 2

Reviewer 1 Report

Comments and Suggestions for Authors

I approve the publishing of this paper.